# Small Intestinal Bacterial Overgrowth (SIBO) and Twelve Groups of Related Diseases—Current State of Knowledge

**DOI:** 10.3390/biomedicines12051030

**Published:** 2024-05-07

**Authors:** Paulina Roszkowska, Emilia Klimczak, Ewa Ostrycharz, Aleksandra Rączka, Iwona Wojciechowska-Koszko, Andrzej Dybus, Yeong-Hsiang Cheng, Yu-Hsiang Yu, Szymon Mazgaj, Beata Hukowska-Szematowicz

**Affiliations:** 1Department of Diagnostic Immunology, Pomeranian Medical University, st. Powstańców Wielkopolskich 72, 70-111 Szczecin, Poland; paulina.roszkowska@pum.edu.pl (P.R.); iwona.wojciechowska.koszko@pum.edu.pl (I.W.-K.); 2Institute of Biology, University of Szczecin, st. Z. Felczaka 3c, 71-412 Szczecin, Poland; klimczak-emilia@wp.pl (E.K.); ewa.ostrycharz@usz.edu.pl (E.O.); szymon.p.mazgaj@gmail.com (S.M.); 3Doctoral School, University of Szczecin, st. A. Mickiewicz 16, 71-412 Szczecin, Poland; 4Molecular Biology and Biotechnology Center, University of Szczecin, st. Wąska 13, 71-412 Szczecin, Poland; 5Department of Genetics, West Pomeranian University of Technology, st. Aleja Piastów 45, 70-311 Szczecin, Poland; aleksandra.raczka@zut.edu.pl (A.R.); andrzej.dybus@zut.edu.pl (A.D.); 6Department of Biotechnology and Animal Science, National Ilan University, Yilan 26047, Taiwan; yhcheng@ems.niu.edu.tw (Y.-H.C.); yuyh@niu.edu.tw (Y.-H.Y.)

**Keywords:** gut microbiota, small intestinal bacteria overgrowth, SIBO, diet, dysbiosis, microbial ecology

## Abstract

The human gut microbiota creates a complex microbial ecosystem, characterized by its high population density, wide diversity, and complex interactions. Any imbalance of the intestinal microbiome, whether qualitative or quantitative, may have serious consequences for human health, including small intestinal bacterial overgrowth (SIBO). SIBO is defined as an increase in the number of bacteria (10^3^–10^5^ CFU/mL), an alteration in the bacterial composition, or both in the small intestine. The PubMed, Science Direct, Web of Science, EMBASE, and Medline databases were searched for studies on SIBO and related diseases. These diseases were divided into 12 groups: (1) gastrointestinal disorders; (2) autoimmune disease; (3) cardiovascular system disease; (4) metabolic disease; (5) endocrine disorders; (6) nephrological disorders; (7) dermatological diseases; (8) neurological diseases (9); developmental disorders; (10) mental disorders; (11) genetic diseases; and (12) gastrointestinal cancer. The purpose of this comprehensive review is to present the current state of knowledge on the relationships between SIBO and these 12 disease groups, taking into account risk factors and the causal context. This review fills the evidence gap on SIBO and presents a biological–medical approach to the problem, clearly showing the groups and diseases having a proven relationship with SIBO, as well as indicating groups within which research should continue to be expanded.

## 1. Introduction

The gut microbiota is now considered a key element in regulating host health. It has been estimated to comprise about 10^14^ cells—more than ten times the number of cells in the human body. It is one of the most diverse ecosystems, being individual and as unique as a fingerprint, and may include up to approximately 1500 different species [1,2]. In recent years, thanks to significant advances in diagnostic technology, knowledge regarding the role of the intestinal microbiota in the human body—which, as is now known, is not limited to the processes of digestion and absorption of nutrients but performs a wide variety of functions—has significantly expanded. The intestinal bacteria affect the various organs of the entire body, including the central nervous system. They can even modulate the body’s behavior through synthesizing various chemical compounds such as serotonin and precursors of neurotransmitters, as well as affecting metabolism and immunity, which play vital roles in the treatment of many diseases [2,3]. The correct quantitative and qualitative composition of the intestinal microbiota—referred to as eubiosis—alters the maintenance of hemostasis and, thus, the health of the entire human body, while dysbiosis (i.e., quantitative and/or qualitative disruptions of the bacterial balance) can cause or exacerbate the course of various pathologies [4]. Abnormal translocation of the intestinal microbiota from the large intestine to the small intestine can result in small intestinal bacterial overgrowth syndrome, abbreviated as SIBO [5,6,7,8]. The excessive proliferation of carbohydrate-fermenting bacteria leads to gas production in the small intestine, resulting in the accumulation of carbohydrates and other products of bacterial metabolism, causing a wide range of ailments beyond the gastrointestinal tract [5,6,7,8].

The aim of this comprehensive review is to present the current state of knowledge on the relationship between SIBO and diseases (divided into 12 groups), taking into account risk factors and the causal context. This topic is justified, as while gastrologists have talked about SIBO for a few years, other medicinal specialists may not have necessarily heard of it. As shown through the review, knowledge about SIBO can help gastroenterologists and specialists in other fields of medicine when making diagnoses; for example, diabetologists can point out that metabolic diseases (e.g., diabetes) may predispose individuals to the development of SIBO. Awareness and knowledge about this relatively new disease can change the lives of people affected by this problem for the better. Even though most symptoms of SIBO may be limited to the gastrointestinal tract, an increasing body of evidence has emphasized the relationships between SIBO and other diseases. For this study, these associated diseases were divided into 12 groups: (1) gastrointestinal disorders; (2) autoimmune disease; (3) cardiovascular system disease; (4) metabolic disease; (5) endocrine disorders; (6) nephrological disorders; (7) dermatological diseases; (8) neurological diseases; (9) developmental disorders; (10) mental disorders; (11) genetic diseases; and (12) gastrointestinal cancer.

This review fills the evidence gap on SIBO. It presents a biological–medical approach to the problem, clearly shows the groups and diseases with a proven strong relationship with SIBO, and indicates groups within which research should be continued.

## 2. Gut Microbiota

The gastrointestinal tract is the second-largest human body system after the cardiovascular system. The small intestine is about 6–7 m long and is the longest part of the digestive tract (with a total length of about 8–9 m), consisting of the duodenum, jejunum, and ileum [9,10]. The presence of numerous pits, folds, crypts, and villi markedly increases the absorptive surface of the small intestine. The intestinal villi are covered with capillary epithelial tissue composed of individual hairs that produce digestive enzymes. Due to their structure, these so-called microvilli increase the absorptive surface of the small intestine and are primarily responsible for absorption of the products of digestion of carbohydrates and proteins from the gastrointestinal tract into the capillaries [1]. After absorption of micro- and macronutrients from the small intestine, thanks to the peristaltic movements of the microvilli, food residues move to the last section of the digestive tract—the large intestine—where mineral salts, vitamins, and some drugs are absorbed, and water is recovered, resulting in thickening of the digestive contents. Moreover, vitamin K and folic acid synthesis are possible due to the abundant bacterial flora, consisting of species such as *Escherichia coli* and *Enterobacter aerogenes* [1]. Pre-digested, still-liquid food content enters from the last segment of the small intestine—the ileum—into the cecum through the ileocecal valve, the mechanism of which ensures the irreversible, unidirectional transport of food debris. It also prevents the transmission of elements of the large intestinal microbiota into the small intestine, which could adversely result in the accumulation of harmful products from bacterial fermentation, such as hydrogen, carbon dioxide, and methane [1,11].

From birth to death, the human body is colonized by many microorganisms—including bacteria, viruses, and fungi, which are commensals, symbionts, and species with pathogenic potential—the collection of which is called the microbiota [12]. The term microbiome was first used in 2001 by American geneticist and microbiologist Joshua Lederberg, a Nobel Prize winner (in 1958 in physiology and medicine, for his discovery of the mechanisms of genetic recombination in bacteria), to describe the genome of all microorganisms inhabiting the human body [13]. The advancement of molecular techniques at the end of the 20th century and their participation in the isolation and identification of microorganisms significantly influenced the development of knowledge about the composition and importance of the human microbiome. It has been shown that the number of microorganisms residing in the human body exceeds the number of cells making it up by more than ten times. In particular, most microorganisms colonize the gastrointestinal tract, and it has been estimated that about 2 kg of microorganisms are present in the large intestine [11].

The microbiota of the gastrointestinal tract significantly influences the immune function of the entire body, the processing of nutrients, and many other significant vital processes that facilitate the maintenance of psychosomatic health. It causes the breakdown of food residues through fermentation, promoting the formation of essential B vitamins, vitamin K, and short-chain fatty acids (SCFAs)—an energy source for colon epithelial cells [14]. Thanks to the microorganisms of the intestinal microbiota, minerals and electrolytes (e.g., sodium, magnesium, calcium, and potassium) are better absorbed. Furthermore, the production of hydrolases affects the metabolism of fats in the liver and, thus, the metabolism of cholesterol and fatty acids. In addition, competition for habitat and nutrients leads to the production of bacteriocins, which prevent the growth of potentially pathogenic bacteria [11].

Microorganisms form a diverse ecosystem that is dynamically modified throughout an individual’s lifetime under the influence of many factors, including diet, antibiotics/chemotherapeutics used, age, lifestyle, social and economic conditions, stress, and metabolites produced by bacterial cells. The oral cavity has an abundant microbiota, which is made up of about 700 different species of bacteria. In contrast, the upper gastrointestinal tract has much fewer microorganisms due to the rapid flow of food content and the low pH of the stomach, as well as the secretory properties of the stomach and duodenum. The number of bacteria in the stomach is about 10 colony-forming units (CFUs)/g of food content, while that in the duodenum is 10^1^–10^9^ CFU/g. Going down the gastrointestinal tract, the number of resident microorganisms gradually increases and their profile changes. In the jejunum, there are about 10^5^–10^7^ CFU/g, dominated by *Bacteroides, Lactobacillus*, and *Streptococcus* species. In the ileum, the number reaches as much as 10^7^–10^8^, dominated by species of the genera *Bacteroides*, *Clostridium*, *Enterococcus*, *Lactobacillus*, and *Veilonella*, as well as those of the order *Enterobacterales* [2,15,16]. Meanwhile, the slower passage of food content in the large intestine favors the existence of microorganisms. Their number is 10^10^–10^14^ CFU/g of gastrointestinal contents, representing about 800 species. These include symbiotic, opportunistic, and pathogenic microorganisms. Anaerobic and relatively anaerobic species of the genera *Bacillus*, *Bacteroides*, *Clostridium*, *Bifidobacterium*, *Enterococcus*, *Eubacterium*, *Fusobacterium*, *Peptostreptococcus*, and *Ruminococcus,* are predominant [2,15,16].

SIBO manifests not only through quantitative changes in microorganisms, but also through qualitative changes [17,18]. The threshold value for the number of bacterial colonies in the diagnosis of SIBO remains undetermined and disputed, as some investigators have considered 10^3^ CFU/mL to be an appropriate cut-off value, while others have considered ≥ 10^5^ CFU/mL. This is due to the limitations of culturing aspirates from the small intestine, the possibility of contamination with flora from the pharynx or oral cavity, and the possibility of false negative results [19].

Mucosa-associated bacterial taxa related to SIBO have recently been identified which may serve as potential biomarkers or therapeutic targets for SIBO (Figure 1), constituting progress in understanding this disease. A comprehensive profile of the gut microbiota in patients with SIBO has been presented by Li et al. [20]. Dysbiosis was observed in the mucosa-associated gut microbiome of patients with SIBO, but not in the fecal microbiome. Different gastrointestinal tract sections in SIBO+ and SIBO− patients were found to be colonized by different bacteria (Figure 1). During the analyses, spectra of microflora from the mucosa of the duodenum, ileum, and sigmoid colon were examined and compared using 16S rRNA gene sequencing. Intestinal mucosal dysbiosis was demonstrated in bacterial overgrowth patients (SIBO+) compared to the control group (SIBO−). In addition, significantly lower species diversity was observed in SIBO+ patients [20].

## 3. Methodology

Electronic databases of scientific publications (i.e., PubMed, Science Direct, Web of Science, EMBASE, Medline) were searched, in which articles in English and Polish from 1991 to 2023 were included. The search terms were as follows: small intestinal bacteria overgrowth, SIBO, gut microbiota, breath test, glucose breath testing (GBT), lactulose breath testing (LBT), dysbiosis, gut dysbiosis, microbial ecology, intestinal microflora abnormalities, and intestinal microbiome. The search results were imported into ENDNOTE 2.0 for the removal of duplicates and selection of publications. The articles were then read, and data related to diseases, studies, methods, participants, and results were collected. Then, the retrieved diseases were assigned into 12 groups. Any disagreements were resolved through discussion between authors.

## 4. Small Intestinal Bacterial Overgrowth (SIBO)

### 4.1. General Characteristics

Small intestinal bacterial overgrowth (SIBO) is defined as a clinical condition caused by excessive numbers of small intestinal bacteria, predominantly including Gram-negative aerobic and anaerobic species [21]. In the physiological state, there are mechanisms to prevent excessive colonization of bacteria in the small intestine, such as acidic stomach pH, pancreatic enzymes, the intestinal immune system, small intestine peristalsis, the ileocecal valve, and the intestinal barrier itself. However, when changes in any of these mechanisms occur, SIBO can develop [22]. Excessive proliferation of carbohydrate-fermenting bacteria leading to gas production in the small intestine can cause the accumulation of carbohydrates, as well as the accumulation of other products of bacterial metabolism, causing discomfort [5,7].

The spectrum of SIBO symptoms includes not only those affecting the digestive tract (e.g., chronic watery/fatty diarrhea, bloating, abdominal pain, constipation, absorption disorders, malnutrition, weight loss, inflammatory changes in the intestines, and atrophy of intestinal villi) but also headaches, mood changes, general malaise, vitamin deficiencies (B12, B1, B3), increased levels of vitamin K and folic acid, D-lactic acidosis, skin symptoms, changes in the liver, and arthralgia [1,5,7,21]. The leading cause of SIBO is a dysfunction in the movement of food content through the small intestine, delayed orocecal transit time (OCTT), and elevated gastric pH; for example, due to prolonged intake of proton pump inhibitors (PPIs) or after gastric surgery.

Small bowel bacterial overgrowth was first postulated as a cause of gastrointestinal symptoms when scientists [23] reported macrocytic anemia in a patient with intestinal strictures. Later scientific reports on SIBO date back only to the 1980s, before which little attention was paid to analyzing the flora of the upper gastrointestinal tract, as it was believed to be sparse and mainly anaerobic. This was also due to the inaccessibility of this area of the gastrointestinal tract for study, as well as methodological and diagnostic limitations. However, with the development of microbiological diagnostic techniques, the microflora of the small intestine has been precisely defined, and its impact on health and disease states is better understood. The growth of many bacteria in the small intestine has been associated with severe metabolic consequences, particularly fatty stools, diarrhea, anemia, and weight loss, even in children [24,25,26,27]. Numerous studies conducted over the past decade have shown that these are just a few of the disease entities directly associated with SIBO.

### 4.2. Diagnostics

Initially, SIBO was identified and diagnosed along with other gastrointestinal abnormalities, including post-operative lesions. However, due to its non-specificity and often asymptomatic course, finding the correct diagnostic technique has been challenging for clinicians and diagnosticians. At present, we can divide the methods used into two groups: invasive (which is the gold standard), including culture from a small bowel biopsy taken during endoscopy; and non-invasive, including breath testing for the presence of hydrogen (H_2_) and methane (CH_4_) after administration of glucose or lactulose (GBT or LBT, respectively) [5,28,29].

The latter can be readily used due to its low cost, general availability, and ease of performance and assay, which is of great importance (especially among pediatric patients). The test involves the administration of 10 g of lactulose or 75 g of glucose, which are substrates for intestinal bacteria that ferment carbohydrates with the production of gas. The patient then exhales every 20 min for 3 h into a breath analyzer that detects the presence of hydrogen or methane. Physiologically, glucose is absorbed in the small intestine (in its proximal part). However, it is fermented during bacterial overgrowth, resulting in gas. These gases are quickly eliminated; however, about 20% goes into the circulation and is absorbed by the lungs and exhaled [5]. For SIBO, the breath test is considered positive when the hydrogen (H_2_) level rises above 20 ppm from baseline for 90 min and the methane (CH_4_) level is ≥10 ppm at any time within 2 h [30,31]. The second criterion proving the presence of small bowel bacterial proliferation is the double peak, which consists of an initial hydrogen peak before 90 min, then a decrease of more than 5 ppm in two consecutive samples, followed by a second hydrogen peak when the substrate enters the cecum [32]. Patients are advised to prepare adequately for the test, which poses the most significant obstacle. It is essential to avoid antibiotics four weeks before the test and drugs that accelerate intestinal peristalsis and have a laxative effect in the week before the test. In addition, no complex carbohydrates or alcohol should be consumed the day before, no food should be consumed 8–12 h before the test, and smoking or exercise should be avoided. Brushing the teeth and rinsing the throat before the test can minimize lactulose fermentation by bacteria in the mouth [32]. Breath tests are characterized by varying sensitivity (52–63% for GBT, 31–68% for LBT) and specificity (82–86% for GBT, 44–100% for LBT) [33].

The gold standard for diagnosing SIBO is culturing of jejunal aspirates. An endoscope is inserted into the second/third part of the duodenum. A Liguora catheter with a valve is placed through the biopsy channel, and about 3 mL of fluid is aspirated with a syringe and immediately sent to a microbiology laboratory for the culture of aerobic and anaerobic bacteria. A bacterial concentration of >10^3^ CFU/mL indicates SIBO [21,29,30]. In addition to its invasiveness, this direct method has other limitations: it is expensive, it requires specialized personnel, and there is a risk of contamination with pharyngeal and oral flora, which can result in false positives. There is no obvious cutoff point to determine a positive aspirate. Moreover, the standard processing of material in the microbiology laboratory does not allow for the detection of all gut microbiome species.

New molecular techniques based on sequencing of the 16S ribosomal RNA gene (which is present in all bacteria) and metagenomic approaches have recently been introduced to overcome the shortcomings of current SIBO testing methods. This is expected to lead to the discovery of new bacterial species, as well as a better understanding of their involvement in SIBO and the associated impacts on related diseases [5].

### 4.3. Treatment and Diet

Treatment of SIBO should be comprehensive, individualized, and if possible, causal (Figure 2). Comprehensive action includes eliminating the underlying disease (e.g., anatomical defect), eradicating bacterial overgrowth through appropriate antibiotic therapy, and making dietary alterations to eliminate nutritional deficiencies [21,33,34]. The former includes surgical management to correct anatomical defects, if any (e.g., adhesions, diverticulosis, intestinal obstruction, fistulas, strictures), as well as elimination or dose reduction, shortening the duration of administration of drugs that reduce intestinal motility or gastric juice acidity (e.g., PPIs), which may promote and exacerbate bacterial overgrowth. In addition, there have been attempts to use prokinetic drugs—which accelerate intestinal motility—in justified cases (e.g., in chronic pseudo-obstruction of the intestines); for example, metoclopramide and erythromycin have been used in the U.S., while in Europe, prucalopride has been used [35]. Due to the difficulty of proper specimen collection, antibiotic therapy for SIBO usually involves empirical treatment using metronidazole, ciprofloxacin, tetracycline, amoxicillin–clavulanate, neomycin, or rifaximin [36,37,38]. Rifaximin is effective in treating SIBO, despite the heterogeneity found in studies and the lack of a recommendation regarding the dose and duration of treatment [38]. SIBO can be recurrent and is statistically more common in the elderly, in patients permanently taking PPIs, and after surgical removal of the appendix, in which case repeated antibiotic therapy and consideration of causal treatment are required [39,40,41].

Increasing incidences of therapeutic failure have prompted the search for novel treatments. Alternative treatments for SIBO include fecal microbiota transplantation from a healthy donor, especially in cases of recurrence or resistance of bacterial strains to commonly used antibiotics. This method can effectively treat certain chronic inflammatory conditions of the gastrointestinal tract, is non-invasive, and does not cause rejection or induce an immune response, frequently allowing for permanent restoration of the normal gastrointestinal microbiota. It involves the oral administration of a “gut microbiota capsule”—a standardized preparation containing previously frozen fecal microbiota—once a week for four weeks [42]. In addition, the positive effects of probiotics, therapeutic diets, and herbal preparations with antimicrobial activity have been reported; however, these modalities are currently only supportive, and confirmation of their efficacy requires further clinical studies [43,44]. The severity of SIBO symptoms has been reduced using probiotics containing various microorganisms [45].

The diet supports the treatment of SIBO, especially in people who have experienced weight loss and vitamin and mineral deficiencies [46]. No restrictive dietary change is required, only periodic avoidance of certain foods. This usually requires the elimination of lactose and other products with a high content of carbohydrates that are not wholly digested or absorbed in the intestines—which thus easily ferment and serve as substrates for intestinal bacteria—and an increase in the coverage of energy needs with fat and the administration of medium-chain triglycerides. The diet is called the low FODMAP diet, and the name derives from the first letters of the carbohydrates to be avoided, including oligosaccharides (e.g., fructans and galacto-oligosaccharides), disaccharides (e.g., lactose), monosaccharides (e.g., fructose), and polyols (e.g., sorbitol and mannitol) [46,47]. This diet was initially designed to alleviate the symptoms of irritable bowel syndrome (IBS), but up to 78% of IBS cases are accompanied by SIBO [48]. SIBO is one manifestation of gut microbiome dysbiosis and is highly prevalent in IBS. The effectiveness of the low FODMAP diet has been assessed primarily in patients with IBS, but the associated data support its effectiveness in SIBO as well, which is why it has been increasingly recommended by doctors [46,48,49]. The diet can positively affect both the resolution of symptoms and prevention of the recurrence of SIBO.

## 5. SIBO-Related Diseases

The symptoms of SIBO may be limited to the gastrointestinal tract; however, a growing number of hypotheses have emphasized the association of SIBO with other diseases. The diseases retrieved during the literature review were divided into 12 groups (Figure 3). Diseases were described within the groups. A descriptive summary is provided under each group. In addition, Table 1 (at the end of Section 5) presents a summary of the findings under each group.

### 5.1. Group 1: Gastrointestinal Disorders

Dysfunctions in the gastrointestinal tract leading to damage to protective barriers can result in the development of SIBO. Risk factors for the development of SIBO include hypochlorhydria (i.e., reduced gastric juice secretion, which occurs physiologically in the elderly), prolonged PPI use, and/or colonization with *Helicobacter pylori* (*H. pylori*) leading to dysbiosis in the stomach and small intestine. An increased incidence of SIBO has been observed in patients infected with *H. pylori* [50,51,52,53,54]. Pelvic and abdominal surgical procedures are a predisposing factor for excessive intestinal bacterial proliferation; notably, gastrectomy results in reduced gastric acid production [55]. It has been estimated that up to 80% of patients suffer from SIBO after bariatric surgery, in an era when this method has become widespread in the treatment of obesity [56].

#### 5.1.1. Irritable Bowel Syndrome (IBS)

The symptoms of SIBO very often resemble those of irritable bowel syndrome (IBS). In recent years, studies have found that the rate of positive SIBO in IBS patients has significantly increased, and it has been found that eradicating overgrown small intestinal bacteria can improve the symptoms experienced by IBS patients. Therefore, the relationship between SIBO and IBS has attracted widespread attention. IBS is the most common functional digestive disorder. In developed countries, its prevalence varies between 10% and 15% [57,58]. It has been estimated that the incidence of bacterial proliferation in the small intestine of patients with IBS was 51.7%, compared to 16.7% in healthy subjects. In addition, patients with IBS and SIBO have more severe gastrointestinal symptoms, likely due to the composition of the intestinal microbiota in SIBO+ and IBS+ patients. In particular, it is dominated by species of the genus Prevotella, which produce enzymes responsible for the fermentation of carbohydrates, through which not only hydrogen and methane can be formed, but also short-chain fatty acids (which are responsible for the hypersensitivity of internal organs) and lipopolysaccharides (LPSs), which, when present in the cell wall, can further induce inflammation of enterocytes [59]. Activation of the immune system and prolonged inflammation in the intestinal mucosa in the course of IBS increase intestinal vascular permeability, while SIBO also induces immune system activity [60]; hence, many studies have confirmed the co-occurrence of SIBO and IBS [61,62,63,64,65,66,67]. Rifaximin therapy caused remission of IBS, indicating SIBO as an inducer of IBS [68,69,70].

#### 5.1.2. Crohn’s Disease (CD) and Ulcerative Colitis (UC)

In other inflammatory bowel conditions, an increased incidence of SIBO co-occurrence has also been observed. Crohn’s disease (CD), a type of inflammatory bowel disease (IBD), is a persistent and irreversible inflammatory disorder with an unknown cause that has a usual state marked by remission and relapse. Frequent fistulas, strictures sometimes requiring surgery, and resection of the ileocecal valve in CD can affect the translocation of the intestinal microbiota [71]. SIBO seems to be a common condition in patients with CD, with an estimated prevalence between 25% and 88%, and is predominant in patients with gastrocolic or jejunocolic fistulas, stasis of intestinal contents, colo-ileal reflux caused by loss of the ileocecal valve, surgical blind loop, intestinal obstruction, and different types of motility disorders [72,73]. SIBO in patients with CD was associated with more severe disease and significant changes in the gut microbiome, which may worsen the symptoms and course of the disease [71,74]. Ulcerative colitis (UC), a non-specific inflammatory condition, is a chronic inflammatory disease of the rectum and colon for which the pathogenesis has not yet been elucidated [75]. In addition, the secretion of pro-inflammatory cytokines in UC can result in SIBO. It has been observed that the eradication of SIBO using rifaximin reduced symptoms in both CD and UC patients [76,77]. UC is characterized by chronic inflammation and repeated attacks. Thus, UC is challenging to cure completely, and it usually presents with clinical symptoms such as fever, bloody diarrhea, abdominal pain, weight loss, and so on [78]. According to one study, patients with IBD—particularly those with CD and female gender who had undergone surgery for IBD—had a higher risk of SIBO than healthy controls [69]. Patients with UC easily experience SIBO, which increases blood endotoxin, TLR2, and TLR4 levels. The synergistic effects of endotoxins and endotoxin receptors TLR2 and TLR4 over-expression mediate body inflammation and may be involved in the progression of UC. UC patients with excessive growth of small intestinal bacteria are more likely to have hypertoxemia [69,79].

#### 5.1.3. Celiac Disease (CeD)

Celiac disease (CeD) is an autoimmune enteropathy triggered by gluten ingestion in genetically susceptible individuals [80]. The link between SIBO and CeD seems quite apparent: intestinal dysbiosis can affect gastrointestinal motility and vice versa [81]. Such a correlation was observed in an earlier study [82]. Patients with CeD present a decrease in cholecystokinin and a sporadic increase in neurotensin; these hormones positively and negatively influence gastrointestinal motility, respectively [83,84]. In addition, rifaximin therapy caused remission of symptoms in patients unresponsive to treatment with a gluten-free diet. However, other later studies [85,86] have shown a lack of response to antibiotic therapy and a lower incidence of SIBO in patients with CeD. CeD has a complex pathogenesis; hence, only specific variants may correlate with SIBO [80,87]. Patients with CeD usually improve on a gluten-free diet (GFD); however, 7–30% of patients continue to have symptoms of malabsorption despite adherence to the GFD. The lack of response to a prescribed GFD or recurrence of gastrointestinal symptoms despite GFD maintenance in patients who responded initially to GFD is usually termed “unresponsive CeD.” Unresponsiveness suggests gluten contamination or the coexistence of other conditions, such as SIBO [80].

#### 5.1.4. Non-Alcoholic Fatty Liver Disease (NAFLD)

Non-alcoholic fatty liver disease (NAFLD) is a chronic liver disease associated with the pathological accumulation of lipids inside hepatocytes. The estimated frequency of NAFLD has been reported to be 25% globally. Untreated NAFLD can progress to non-alcoholic hepatitis (NASH), followed by fibrosis, cirrhosis, and hepatocellular carcinoma [88].

Increasing evidence supports the association between SIBO and NAFLD [88,89,90,91]. The relationship between these two diseases is further intensified, as both are linked by signaling pathways, creating the gut–liver axis [89]. The health consequences of SIBO are primarily malabsorption disorders (vitamin B12, iron, choline, fats, carbohydrates, and proteins) and deconjugation of bile salts [88,89,90]. Undetected and untreated SIBO can lead to nutritional and energy malnutrition, directly impairing liver function [89]. Numerous studies have indicated the frequent occurrence of SIBO in patients with NAFLD, including children. Rafiei et al. [92] examined 98 Iranian patients with NAFLD for SIBO using a hydrogen breath test. SIBO was demonstrated in 38 patients (39%), and SIBO+ patients reported flatulence, while SIBO+- patients reported abdominal pain. Gkolfakis et al. [93] showed that the incidence of SIBO was significantly higher in a cohort of NAFLD patients compared to the control group. Moreover, SIBO is more common in patients with NASH-related cirrhosis compared to patients with NAFLD. Other studies [94] have shown that SIBO is associated with NAFLD and SIBO+ patients are at increased risk of fatty liver disease. In a study conducted in a large cohort of 372 patients, 141 (37.9%) were SIBO+, while 231 (62.1%) obtained a negative result and thus constituted the control group. In turn, Shi et al. [95] showed that patients with NAFLD have an increased incidence of SIBO and prolonged OCTT. SIBO was diagnosed using LBT. SIBO in patients with NAFLD may be a factor contributing to increased transaminase activity, hepatic steatosis, progression of liver fibrosis, and prolonged OCTT. Their study included 103 patients with NAFLD and 49 healthy people (control group). In turn, Mikolasevičal [96] showed that SIBO was more common in patients with NAFLD and significant fibrosis than in patients without NAFLD and without fibrosis. Moreover, it was demonstrated that significant predictors associated with SIBO were the degree of fibrosis, steatosis, and degree of ballooning, as well as Gram-negative bacteria, particularly *Escherichia coli* and *Klebsiella pneumonia*. NAFLD was diagnosed in 117 patients using Fibroscan with a controlled attenuation parameter, as well as liver biopsy. SIBO was defined through esophagogastroduodenoscopy with aspiration of the descending duodenum.

The association of NAFLD with SIBO in obese and overweight children has also been studied [97,98,99]. A meta-analysis [100] was conducted based on the three previously mentioned studies, involving 205 children, which indicated a possible relationship between SIBO and NAFLD in children. SIBO+ children were more likely to have NAFLD, and children with NAFLD had an over 2-fold increased relative risk of developing SIBO. Other studies have indicated that SIBO may be a pathophysiological factor in the development and progression of NAFLD, altering intestinal permeability and allowing bacterial endotoxins to enter circulation [101]. Miele et al. [102], in a group of 35 patients with NAFLD, examined intestinal permeability and assessed the correlation between this phenomenon and the disease, the integrity of tight junctions in the small intestine, and the presence of SIBO. SIBO was diagnosed using GBT. NAFLD in humans is associated with increased intestinal permeability, and this abnormality is associated with an increased incidence of SIBO in these patients. On the other hand, another study [103] has reported that serum endotoxin levels did not differ between SIBO+ and SIBO− patients, nor did they change after antibacterial therapy, which practically excludes the possibility that increased endotoxemia in patients with NAFLD without cirrhosis liver disease is associated with SIBO.

#### 5.1.5. Liver Cirrhosis

SIBO often appears in people with liver cirrhosis [104,105], and bacterial overgrowth was found in as many as 51.5% of patients during these tests. A meta-analysis of 21 studies revealed that the prevalence of SIBO in patients with cirrhosis was greater than that in healthy controls [106]. The disease affects the liver and other organs, including the intestine [107]. This condition is caused by impaired intestinal motility, delayed intestinal transit, pancreatic exocrine insufficiency, and/or immunological disorders. Additionally, patients with this type of liver disorder suffer from comorbidities such as diabetes or autonomic neuropathy, which intensify the occurrence of SIBO [107,108]. Liver cirrhosis causes dysbiosis; there is an increase in the number of bacteria of the Proteobacteria genus, which are responsible for the production of active endotoxin, and anaerobic bacteria of the Bacilli genus, which can translocate bacteria. There is also a decrease in the level of *Clostridium* bacteria. SIBO patients mainly have bacteria from the Blautia genus, which can convert primary bile acids into secondary bile acids. When changes in bile metabolism occur, the intestinal microbiome changes; the more bacteria there are, the more deconjugation of primary fatty acids in the small intestine will increase. These acids, however, have a lower affinity for proteins, thus causing them to be transported across the epithelium of the terminal ileum [107]. Meta-analyses have shown that SIBO in cirrhosis is associated with hepatic encephalopathy [109].

#### 5.1.6. Pancreatitis

There have also been reports of a higher incidence of SIBO in patients with chronic as well as acute pancreatitis; however, only small groups of patients were studied [110,111,112,113,114]. SIBO can complicate chronic pancreatitis and interfere with its management [115]. It has been demonstrated that intestinal barrier dysfunction allows bacteria of intestinal origin to transfer to extraintestinal organs, leading to sepsis and subsequent infectious diseases with high mortality rates [116]. According to multiple studies, the presence of SIBO in acute pancreatitis correlates with the severity of the disease [116,117]. SIBO is often present in patients with chronic pancreatitis with persistent steatorrhea despite pancreatic enzyme replacement therapy. The overall prevalence of SIBO diagnosed after GBT varies between 0% and 40% but is 0–21% in those without upper gastrointestinal surgery [115,117].

**Group 1 summary:** In summary, it should be stated that group 1 best describes the relationship between SIBO and related diseases. In this group, SIBO occurs with high incidence in patients with CD, liver cirrhosis, IBS, NADFL, and pancreatitis. In the case of NAFLD, significant predictors associated with SIBO are the degree of fibrosis, steatosis, and degree of ballooning, as well as the presence of Gram-negative bacteria, which gastroenterologists and other specialists should consider when diagnosing. In turn, the association between SIBO and CeD requires further research, as CeD has a complex pathogenesis; hence, only specific variants may correlate with SIBO.

### 5.2. Group 2: Autoimmune Disease

#### Systemic Sclerosis (SSc)

Systemic sclerosis (SSc) is a chronic connective tissue disease of autoimmune origin leading to organ fibrosis. The pathological process also affects the gastrointestinal tract [118,119,120]. About 55% of patients suffer from gastrointestinal complaints (e.g., bloating, abdominal pain, constipation, diarrhea). SIBO occurs with a prevalence of approximately 39–62% of patients with SSC and presents with non-specific gastrointestinal tract symptoms [121,122,123,124]. Abnormal intestinal motility, prolonged OCTT, and impaired intestinal clearance are factors affecting the development of SIBO. Several studies have confirmed the co-occurrence of SIBO and SSc, and a disease duration of more than five years is a significant risk factor for the development of SIBO [35,123,124,125,126,127].

The data reveal a strong link between SIBO and SSc, with a 10-fold increased prevalence of SIBO in SSc patients compared to controls [125]. Moreover, the risk of diarrhea was higher in SSc patients with SIBO than in those without SIBO. Parodi et al. [35] showed that, when comparing 55 SSc patients and 60 healthy controls, the prevalence of SIBO was higher in SSc patients than in controls (30/55 vs. 4/60, respectively). Intestinal symptoms in these patients may be related to this syndrome, and its eradication seems helpful in improving clinical features. OCTT is significantly delayed in SSc patients, suggesting impairment of intestinal motility—a further risk factor for the development of SIBO. Other studies have also shown that velocity through the small intestine is significantly reduced in SSc patients with diffuse abdominal symptoms [126]. A total of 15 individuals with SSc (13 women, median age 58 years; all suffering from diffuse abdominal symptoms) and 17 healthy volunteers (12 women, median age 52) were evaluated in this study using a motility tracking system, which measures gastric emptying and velocity through the small intestine. SSc patients were examined for bacterial overgrowth using the hydrogen breath test and radiopaque markers to determine the total gastrointestinal transit time. A few SSc patients (21%) had positive breath tests for small intestinal bacterial overgrowth. In a large cohort of 99 SSc patients, the most frequently involved organ was the esophagus, followed by the small intestine and stomach [127]. All 99 patients underwent LBT to evaluate for SIBO. LBT was consistent with a diagnosis of SIBO in 47 of 99 SSc patients (46%), compared with 3 of 60 controls (5%). PPI use was associated with a higher incidence of SIBO. Thus, SIBO affected almost half of the SSc patients included in the study [127]. Sawadpanich et al. [123] have evaluated the prevalence and associated factors of SIBO in SSc patients in Thailand. A total of 89 SSc patients (30 male and 59 female, mean age: 54.4) underwent the glucose H_2_/CH_4_ breath test, for which 12 participants obtained positive results (SIBO prevalence of 13.5%). A duration of disease > 5 years was significantly associated with SIBO. Differentiation of the fecal microbiome between patients with SSc and SIBO and without SIBO was noticed by Levin et al. [124]. Their study group included 29 patients with SSc (27 women and two men) and 20 healthy controls. The 29 patients with SSc underwent respiratory testing to evaluate for SIBO, and 13 SSc patients were diagnosed with SIBO (44.8%). Stool samples were compared from patients with SSc, with and without SIBO, and healthy controls (n = 20) aged 18–80. Fecal microbiome analyses demonstrated differences between SSc patients with and without SIBO. For patients with SSc and SIBO, significant differences in bacterial diversity but not richness were found. There was a significantly larger relative abundance of *Bacteroides* spp. and *Uncl. Rickenellaceae* spp. in SIBO+ SSc patients compared to healthy controls. In addition, a significantly smaller relative abundance was found in *Uncl. Erysipelotrichacaea* spp. in SIBO+ SSc patients compared to controls. Fecal microbiome analyses demonstrated species diversity differences between healthy controls and patients with SSc, where SSc patients exhibited higher relative abundances of Proteobacteria and Bacteroidetes and lower abundance of Firmicutes. The Firmicutes/Bacteroidetes ratio was substantially lower in patients with SSc. At the genus level, SSc patients exhibited a lower relative abundance of *Enterococcus* spp. and *Lactococcus* spp., while higher relative abundances of *Bacteroides* and *Lachnospira* were also observed. Current fecal microbiome research in SSc is focused on finding the best candidate for microbiota-targeted therapy to control SIBO [128].

Antibiotic treatment can eradicate SIBO and improve gastrointestinal symptoms in SSc patients [129]. A reduction in gastrointestinal symptoms has been observed following rifaximin therapy, and recent studies have also demonstrated the efficacy of *Saccharomyces boulardii* probiotics in combination with metronidazole in treating SIBO [35,121,130]. Fecal calprotectin (FC) [131] turned out to be a sensitive and specific marker differentiating patients in the control and study groups (SSc detection). As FC levels of ≥275 μg/g are markedly associated with SIBO, these findings suggest that FC determination may be a helpful test in identifying the group of SSc patients at high risk for SIBO, requiring GBT to detect SIBO. Moreover, FC levels may be helpful to assess SIBO eradication in SSc patients, as long-term antibiotic therapy is costly and carries risks such as the onset of pseudo-membranous colitis and SIBO-related antibiotic resistance [131,132]. 

**Group 2 summary:** In summary, it should be noted that the relationship between SIBO and SSc is very well clinically documented. SIBO occurs at a 10-fold higher incidence in SSc patients than in healthy people. Combining this information with the fact that FC in stool is a sensitive and specific marker differentiating patients with and without SSc, physicians and other specialists can use this information in practice.

### 5.3. Group 3: Cardiovascular System Diseases

SIBO is linked to the cardiovascular system. Reduced cardiac output in the course of its failure causes ischemia of the small intestinal wall, which in turn leads to intestinal dysfunction and increased permeability. High levels of LPS—a component of the cell walls of Gram-negative bacteria and a pro-inflammatory factor—have been observed in patients with cardiovascular edema [133]. Endotoxin binds to the TLR-4 receptor and activates inflammation, leading to left ventricular remodeling and increased apoptosis of myocardial cells [134,135].

#### 5.3.1. Heart Failure (HF)

Patients with heart failure (HF), malnutrition, and cachexia often develop a systemic inflammatory and immune response. The cause of this condition is believed to be altered intestinal function, which causes bacteria to translocate and microorganisms and endotoxins to enter the circulation. It has been shown that patients with HF have a more dysfunctional intestinal flora than healthy people. In this regard, metagenomic analyses of the 16S rRNA gene sequence in excrement samples have been performed [136]. Increasing evidence supports the correlation between HF and gut microbiota. In patients with HF, the exhaled concentrations of hydrogen have been related to HF severity and a higher risk of adverse outcomes [137]. Recent studies have demonstrated a bidirectional relationship between the intestinal flora and the brain. The interaction occurs through neural, hormonal, and immune pathways and is called the gut–brain axis. It also occurs in ischemic stroke (AIS); the prevalence of SIBO in a sample of patients with AIS was found to be 28.8% [138].

#### 5.3.2. Deep Vein Thrombosis (DVT)

Deep vein thrombosis (DVT) may also be associated with SIBO. Due to increased levels of inflammatory factors and expression of TLR-4—which, when combined with lipopolysaccharides, enhances procoagulant activity—SIBO is a risk factor for DVT. The cytokine storm in DVT may stimulate intestinal immunity through affecting its microflora and is thus a risk factor for SIBO [139].

#### 5.3.3. Coronary Artery Disease (CAD)

He et al. [140] have suggested that alcohol consumption may be associated with CAD and SIBO. The authors speculated that SIBO+ patients may produce a larger quantity of endogenous alcohol than SIBO− patients, which may lead to the occurrence of fatty liver disease in SIBO+ patients, which was observed more frequently in this group when compared to the control group.

#### 5.3.4. Subclinical Atherosclerosis

Metabolic products generated by gut bacteria have been implicated in the development of subclinical atherosclerotic lesions [141,142]. Humans need gut bacteria to fulfill their vitamin K2 requirement, which cannot be provided through the diet; this is especially true of the Western population. SIBO is associated with impaired human vitamin K metabolism. For this reason, SIBO, low vitamin K2 status, or both could hypothetically pose an increased risk for the development of atherosclerotic disease [142]. It was confirmed through a study conducted in 2022 [141] that SIBO is associated with subclinical atheromatous plaques. Plaques were more common in the SIBO+ group than in the SIBO− group, and the mechanism of this association warrants further exploration [141].

**Group 3 summary:** In summary, it should be noted that knowledge regarding the relationship between SIBO and diseases in this group is flourishing, promoting continued expansion in this field of research. Established facts, such as the relationship between SIBO and HF and the risk of adverse reactions occurring in this system, make it possible for specialists to apply this advice in the fields of gastroenterology and cardiology. Another issue is whether SIBO is a risk factor for DVT, CAD, and subclinical atherosclerosis. The association between SIBO and the latter diseases requires further research, which is only a matter of time.

### 5.4. Group 4: Metabolic Diseases

#### 5.4.1. Diabetes

The incidence of SIBO was significantly higher in patients with both type I and type II diabetes when compared to a healthy population [143,144]. The pathogenesis of SIBO in diabetes is not entirely understood, although it is known to be multi-factorial, involving hyperglycemia, delayed gastric emptying, or impaired gut motility resulting from diabetic autonomic neuropathy [144]. Diabetic neuropathy—resulting from dysfunction of the intrinsic autonomic nerves and the vagus nerve—affects the physiology of the entire gastrointestinal tract, causing (among other things) stasis in the small intestine, a risk factor for SIBO [145]; furthermore, the maintenance of dysbiosis can exacerbate diabetic symptoms [146,147].

SIBO is associated not only with type I and type II diabetes, but also with gestational diabetes mellitus (GDM). In a population of pregnant Chinese women with gestational diabetes, symptoms of bloating abdominal pain were related to higher concentrations of exhaled hydrogen and methane compared to the control group. In addition, the blood glucose content of women with GDM and SIBO was higher when compared to patients with gestational diabetes without symptoms of SIBO; also, the birth weight of newborns was higher in women with SIBO [148]. Treatment with prebiotics, probiotics, and antibiotics reduces gastrointestinal symptoms in patients with diabetes [146,147]. The study found that diabetes patients with autonomic neuropathy have a significantly higher prevalence of SIBO than those without autonomic neuropathy. These results suggest that diabetes mellitus could be a predisposing factor for the development of SIBO [145,149]. Existing results suggest that type II diabetes combined with SIBO is inversely associated with insulin secretion and worse glycemic control [145,149].

#### 5.4.2. Hyperlipidemia

SIBO may cause hyperlipidemia through enterohepatic circulation disturbance, which evolves against the background of early bile acid deconjugation with further endotoxin production and oxidative stress in the liver, leading to hyperproduction of cholesterol and atherogenic lipoproteins. SIBO was present in 78.9% of patients with hyperlipidemia and 40% of control subjects, and relationships between the H_2_ rate and low-density protein, triacylglycerols, and very low-density lipoprotein levels were observed [150].

#### 5.4.3. Obesity

The issue of obesity is becoming more and more involved worldwide, as the number of people with excess weight is increasing every year. The environment, diet, lack of sleep, dysregulated hormonal balance, chronic inflammation and, most importantly, human microbiome contribute greatly to this issue [151]. The composition of the intestinal microflora has been compared between thin individuals and those with obesity. Relevant research has mainly been conducted based on experiments performed on mice [152]. In obese mice, a higher number of bacteria of the Firmicutes genus was found, as well as a lower number of *Bacteroidetes* when compared to lean rodents. As a result of genetic analysis, it was found that a higher level of the latter is responsible for greater energy use from food. When gut bacteria were transplanted into obese germ-free mice from lean mice, it resulted in less weight gain [153].

Obesity is associated with an increased risk of SIBO, despite inconsistent results [154,155,156,157]. The statement has been confirmed in studies on the relationship between SIBO and obesity, where this relationship was demonstrated despite a small sample size (based on an LBT and a study with a wireless motor capsule) [155]. The risk of SIBO was twice as high in patients with obesity compared to the non-obese control group. Furthermore, when only studies from Western countries were considered, the risk was threefold. This is probably due to the use of different diagnostic methods with different sensitivity and specificity [154]. There are several mechanisms influencing the occurrence of obesity, the first of which is the more efficient absorption of energy from food by the intestinal microflora. Studies have shown elevated levels of SCFAs—an energy compound derived from the digestion of carbohydrates in the large intestine—in obese people, as simple carbohydrates are absorbed more, resulting in increased liver lipogenesis and adipose tissue formation [153]. Another factor causing obesity is a change in the motility of the gastrointestinal tract. It was found that patients had slower intestinal transit but greater absorption in its proximal part. Prolonged transit through this part of the digestive system leads to weight gain. The smooth muscles of the small intestine in obese people show greater contractility than those in healthy people, which may be the reason for faster transit in the proximal part of the intestine [158]. The causes of obesity also include accelerated gastric emptying in patients with excess weight, which is related to over-eating. This reduces the reception of the feedback signal related to satiety, resulting in a faster feeling of hunger [159]. Another reason may be that the intestinal microflora is changed in people with obesity. Diet also impacts the microbiome; for example, the Western diet, which is rich in fats and carbohydrates, reduces the level of beneficial microorganisms in the intestines [154].

**Group 4 summary:** Considering the results within this group, we can conclude that a relationship exists between SIBO and metabolic diseases. In this group, SIBO occurs with a high frequency in patients with hyperlipidemia and obesity. In particular, diabetes physicians should pay attention to the fact that SIBO may develop in patients with diabetes, which may result in worse glycemic control.

### 5.5. Group 5: Endocrine Disorder

#### Thyroid Disease

Endocrine disorders are very commonly accompanied by gastrointestinal symptoms in the form of constipation, diarrhea, or abdominal discomfort. Despite very little data and its unexplained pathogenesis, it appears that hypothyroidism can slow down intestinal motility and more than 50% of patients with hypothyroidism also suffer from SIBO [160]. Interestingly, it has been observed that levothyroxine supplementation strongly interferes with SIBO [161]. Hyperthyroidism can also affect gastrointestinal motility, but the causes remain unclear [162]. Bacterial antigens can induce both Hashimoto’s disease and Graves-Basedow’s disease, which have an autoimmune basis. A state of intestinal dysbiosis modifies the immune response, inducing inflammation which in turn increases the sensitivity of the intestinal wall and consequently increases its permeability, provoking the immune system to produce antibodies that may play an essential role in the development of Hashimoto’s or Graves-Basedow’s disease [163]. Good dietary habits and consumption of optimal amounts of omega-3 fatty acids, fiber, and plant products play a significant role in improving the intestinal microbiome, which can reduce inflammation [161,164,165]. In one of the most extensive studies devoted to examining the occurrence of SIBO in patients with hypothyroidism, it was shown that the most important contributors to the development of SIBO are levothyroxine use, impairment of intestinal clearance, and immunosuppression. Still, these factors do not sufficiently explain its emergence [161]. Other studies [164,165] have confirmed that hypothyroidism is associated with the development of bacterial overgrowth. Excess bacteria could influence clinical gastrointestinal manifestations. Bacterial overgrowth decontamination is associated with improved gastrointestinal symptoms. However, fermenting carbohydrate luminal bacteria does not interfere with thyroid hormone levels. In 2021, it was demonstrated that the occurrence of subclinical hypothyroidism is related to SIBO, and the excessive growth of small intestinal bacteria may affect gastrointestinal symptoms [166].

**Group 5 summary:** Although this group is represented only by thyroid disease, due to its frequent occurrence in the human population, it occupies an essential place in this review. Endocrinologists should pay attention to the fact that in patients with hypothyroidism, one of the predictors of the development of SIBO is the use of levothyroxine.

### 5.6. Group 6: Nephrological Disorders

#### 5.6.1. Chronic Kidney Injury

One of the most novel issues of recent interest to many scientists is the relationship between kidney wellbeing and the gastrointestinal tract, that is, the so-called gut–kidney axis [167]. Recent results indicate that the intestinal environment considerably impacts kidney function and vice versa. The majority of patients with chronic kidney disease have gastrointestinal symptoms. Renal failure, in turn, results in a predisposition to changes in the gastrointestinal microbiome, associated with the overgrowth of aerobic and proteolytic bacteria in the duodenum and jejunum [167]. An excessive increase in bacteria of the *Enterobacterales* and *Enterococcus* genera with a concomitant decrease in *Lactobacillus*, *Prevotella*, or *Bifidobacterium* appears to be significant in this respect [167]. In addition, in patients with chronic renal failure, changes in the structure of the intestinal wall have been observed, including degradation of intestinal villi, enlargement of crypts, disruption of epithelial cell integrity, infiltration of inflammatory cells in the lamina propria, and impaired excretion of bacterial metabolites (e.g., indoxyl sulfate and p-cresyl) in the urine, thus promoting intestinal disorders [168,169].

A significantly higher incidence of SIBO has been observed in patients with chronic kidney injury (CKI) compared to controls [170]. The study covered 12 patients with CRF and GI symptoms (ten men, two women) and 10 patients with CRF without GI symptoms (five men, five women). Uremic toxins—whose precursor amino acids (e.g., tryptophan, phenylalanine, and tyrosine) are produced during bacterial metabolism, then converted in the liver to uremic toxins (indoxyl sulfate and p-cresyl sulfate), which are removed under physiological conditions by the kidneys along with urine—seem to be responsible for this mechanism. Their accumulation in patients with chronic renal failure affects the autonomic nervous system of the gastrointestinal tract, worsening intestinal motility and promoting excessive bacterial growth. On the other hand, increased intestinal permeability in CKI patients allows antigens to enter the bloodstream. Then, bacterial toxins can be captured by anion transporters in the renal tubules, impairing their function [171,172].

#### 5.6.2. Acute Kidney Injury

During acute kidney injury (AKI), intestinal dysbiosis may also be observed. Regarding the main initiating factor of AKI—ischemia–reperfusion injury (IRI)—its complex pathogenetic mechanism is potentially related to the formation of reactive oxygen species which damage endothelial cells. The release of inflammatory mediators and the expression of adhesion molecules is therefore initiated during ischemia and subsequent reperfusion, which can induce intestinal dysbiosis and further induce inflammation, thus exacerbating kidney damage [173]. A study conducted by Yang et al. [174] in mice reported the renoprotective effect of reducing intestinal microflora, which was associated with a decrease in the activity of Th1 and Th17 lymphocytes and an increase in the activity of Treg lymphocytes and macrophages. Intestinal dysbiosis is also thought to be a factor in the development of IgA nephropathy [175]. Despite the above reports, no complex data allowing for evaluation of the association between SIBO and kidney disease are available at present.

**Group 6 summary:** In summary, knowledge regarding the relationships between SIBO and diseases in this group is lacking and thus requires continued expansion. Undoubtedly, established facts such as the relationship between SIBO and CKI make it possible to observe patients for SIBO. Determination of the association between SIBO and AKI unquestionably requires further research in a representative cohort of patients.

### 5.7. Group 7. Dermatological Diseases

#### 5.7.1. Acne Rosacea

Acne rosacea is a chronic inflammatory acne-like dermatosis that occurs in the mid-face area and manifests as transient erythema, redness, pustules, and papules [176]. The direct cause is unknown at present; however, the mechanisms affecting this disease may include changes in the immune response [177]. Some cases of acne are associated with various gastrointestinal diseases, including IBS, IBD, CeD, gastroesophageal reflux disease, *Helicobacter pylori* infection, and SIBO. The latter is thought to be influenced by the gut–skin axis [178]. Patients with rosacea were as much as 13 times more exposed to SIBO compared to controls, and regression of skin lesions was observed after treatment of SIBO with rifaximin [177]. The mechanism for the co-occurrence of SIBO and acne rosacea remains unexplained, but it is suspected that SIBO increases intestinal permeability, resulting in the transfer of bacteria and pro-inflammatory cytokines into the systemic circulation, causing skin inflammation through increasing levels of cytokines such as TNF-α, which inhibits interleukin 17 and stimulates a Th1-dependent immune response [177,178]. One study has demonstrated that acne rosacea patients have a significantly higher SIBO prevalence than controls (52/113 vs. 3/60) [179]. After eradication, cutaneous lesions cleared in 20 of 28 and greatly improved in 6 of 28 patients, whereas patients treated with a placebo remained unchanged (18/20) or worsened (2/20). Placebo patients were subsequently switched to rifaximin therapy, and SIBO was eradicated in 17 of 20 cases, with 15 experiencing a complete resolution of rosacea. After antibiotic therapy, 13 of 16 patients with negative BTs for SIBO remained unchanged, and this result differed from SIBO-positive cases. Eradication of SIBO induced an almost complete regression of their cutaneous lesions, and this excellent result was maintained for at least 9 months [179].

#### 5.7.2. Psoriasis

Another dermatological condition, psoriasis, is a chronic inflammatory condition of the skin with a multi-factorial basis manifesting through the formation of scaly, erythematous, and hardened skin plaques, which causes epidermal hyperplasia, proliferation of skin blood vessels, and inflammatory infiltration of T lymphocytes. It affects both children and adults but is more common in the latter [180]. The relationship between psoriasis and SIBO has been described in two studies [181,182], both from 2018. Psoriasis can involve different body parts, such as the oral cavity, in which case it manifests itself as a so-called geographic tongue [181]. It is a benign inflammation of the tongue that “wanders” on its surface, appearing as red, oval-shaped lesions surrounded by a white border. These areas change shape, size, location, and pattern, and there are alternating periods of exacerbation and complete remission of symptoms [183]. The literature has described a case of a woman with lingual psoriasis without psoriatic skin lesions [181]. The woman experienced pain, a burning tongue, and difficulty in eating. In addition, she was diagnosed with SIBO, and after the implementation of rifaximin, a significant improvement in the condition of the tongue was observed, eliminating pain and discomfort. After three months, no recurrence of psoriatic lesions in the oral cavity was observed [181]. The combination of rifaximin with a prebiotic that can benefit intestinal motility seems to be more useful in eradicating SIBO than rifaximin alone. Moreover, rifaximin is similar to rifampin, which is a biofilm-dispersing agent in addition to its antibacterial action. In other studies, Drago et al. [182] have shown that the prevalence of SIBO in patients with psoriasis is comparable to that in the control group. Therefore, further research is needed in order to explain the relationship between psoriasis and SIBO.

**Group 7 summary:** In terms of this group, it was found that the incidence of SIBO in acne rosacea patients is very high, amounting to 46% [6]. Dermatologists should consider this fact, as the eradication of SIBO caused the acne to regress for a long time. In turn, the relationship between SIBO and psoriasis requires further research in a representative cohort of patients.

### 5.8. Group 8. Neurological Diseases

#### 5.8.1. Alzheimer’s Disease (AD)

Alzheimer’s disease is the most common neurodegenerative disease, for which the number of sufferers worldwide has been estimated at 15–21 million. It is a degenerative disease of the brain caused by the deposition of pathological proteins (beta-amyloid, tau protein, and alpha-synuclein) in the brain, causing atrophy of neurons and their connections. This results in a decrease in the amount of transmitter substances needed for normal brain function [184]. The pathogenesis of AD is not clear-cut and is most likely multi-factorial, sometimes showing significant links to other diseases such as depression, diabetes, cardiovascular disease, and IBD [184]. The gut microbiome regulates the immune system and the central nervous system, affecting the adequate production of neuroactive molecules (e.g., serotonin, acetylcholine, tryptophan, histamine, GABA, and catecholamines) [185]. The exact mechanisms of gastrointestinal comorbidities with AD are not yet known. However, studies have suggested that an abnormal structure of the intestinal microbiota results in the production of toxic metabolites that penetrate the bloodstream to the brain and cause inflammation in the nervous system. The stools of AD patients showed a less diverse microbiome and less diverse calprotectin levels with respect to the control group [185]. Few studies have confirmed the co-occurrence of SIBO and AD [186,187]. During a study conducted at the University of Singapore to analyze the prevalence of SIBO, respiratory tests were performed in AD patients, and the results were compared with a sex- and age-matched control group. The frequency of positive breath test results was similar in both groups [186]. Other results from 2022 [187] indicate that the gut microbiota contributes to the pathogenesis of Alzheimer’s disease (AD). The prevalence of SIBO was significantly higher in AD patients than in age-matched controls without dementia (49% vs. 22%) [187].

#### 5.8.2. Parkinson’s Disease (PD)

Parkinson’s disease is the second most common neurodegenerative disorder in the elderly population worldwide. The occurrence of neuropathological changes characterizes PD via the abnormal accumulation of α-synuclein and necrosis of dopaminergic neurons in the black matter [188]. In addition to typical symptoms such as tremors and stiffness of the limb bones, sleep disturbances, constipation, and depression, patients also experience gastrointestinal symptoms resulting from abnormal motility and delayed gastric emptying [188]. *H. pylori* co-infection synergistically exacerbates these gastrointestinal symptoms [189,190]. Numerous studies have confirmed the association between SIBO and Parkinson’s disease [188,189,190,191,192], with as many as 14% to 67% of patients struggling with SIBO depending on the demographic and clinical characteristics of the population included in the study, as well as on the testing method that has been used [193]. Li et al. [188] demonstrated a strong association between SIBO and PD in a group of 973 participants, with approximately half of PD patients testing positive for SIBO. These relationships significantly differed based on the diagnostic test used (GBT, LBT) and geographic area. The prevalence of SIBO was 52% among patients from Western countries and 33% among patients from Eastern countries. Niu et al. [189] tested 182 Chinese patients with PD patients and included 200 sex-, age-, and BMI-matched subjects without PD. SIBO was highly prevalent in PD, with a rate of nearly one-third detected. SIBO was associated with worse gastrointestinal ailments and worse motor function. A similar study [191] has reported that SIBO was detected in a quarter of patients, including those recently diagnosed with the disease. SIBO was not associated with worse gastrointestinal symptoms but independently predicted worse motor function. Appropriately designed clinical trials are needed to confirm a causal relationship between SIBO and poorer motor function in PD. SIBO can trigger a local inflammatory response, disrupting the integrity of the intestinal barrier. Increased intestinal permeability leads to increased exposure of the mucosa to bacterial exotoxins and LPS—factors responsible for local inflammation—which can increase α-synuclein amyloidogenesis and neuronal susceptibility to neurodegeneration. SIBO also affects the absorption of enteral drugs directed to treat PD; gut dysfunction due to microflora abnormalities results in impaired absorption of levodopa, a key drug in the treatment of PD, and consequently reduced dopamine concentrations in the target organ (i.e., the brain) [191,192]. Excessive numbers of bacteria in the gut can induce the production of reactive oxygen species, which inactivate the drug and alter its bioavailability. There is a strong association between SIBO and Parkinson’s disease, primarily because the neurodegenerative disease involves both the autonomic and enteric nervous systems; notably, the vagus nerve is responsible for innervating the stomach, small and large intestine, and the appendix [193]. The hallmarks of Parkinson’s and SIBO are malnutrition and osteoporosis. During bacterial overgrowth, unconjugated bile acids predominate, bile acid synthesis is inhibited, and bile acid levels are reduced, resulting in decreased lipid absorption and weight loss due to fat loss. In addition, the absorption of fat-soluble vitamins (primarily vitamin D) is reduced, which can exacerbate osteoporosis in Parkinson’s disease [188]. Interestingly, the incidence of SIBO does not depend on the duration of PD: it can occur at an early stage with the same frequency. Hence, it has been hypothesized that dysbiosis is not an effect but instead one of the causes of PD, and improving the gut bacterial status may have a positive effect on the manifestation and progression of PD; however, further research is required to validate this hypothesis [191].

#### 5.8.3. Multiple Sclerosis (MS)

Zhang et al. [194] showed that SIBO is highly prevalent in Chinese patients with multiple sclerosis (MS), where 45 of 118 MS patients were SIBO+. Further research is required to establish a causal association between SIBO and the risk and progression of MS.

**Group 8 summary:** In summary, it should be stated that a strong relationship exists between SIBO and neurological diseases. The incidence of SIBO in AD and PD is comparable, amounting to 48% and 46%, respectively [6]. Moreover, it has been suggested that SIBO may cause PD and influence the progression of the disease. In turn, the association between SIBO and MS requires further research, as the retrieved research was only conducted in a small cohort of Chinese patients.

### 5.9. Group 9: Developmental Disorders

#### Autism Spectrum Disorders (ASDs)

Autism spectrum disorders (ASDs) are now increasingly being diagnosed, although their pathogenesis is still unknown. Autism severity is divided into three groups (mild, moderate, severe) and is measured based on the Autism Treatment Scale (ATEC) established by Bernard Rimland and Stephen Edelson of the Autism Research Institute [195]. Several studies in patients with varying degrees of autism have shown significant modifications to the gut microbiota composition. Higher levels of *Bacteroides* or *Clostridium perfringens* were observed in the autistic population, while higher *Firmicutes* were observed in the healthy population [196,197]. Wang et al. [198] showed that children with autism had an SIBO incidence of 31.0%, which was higher in comparison to controls. SIBO+ autistic children had higher autism treatment evaluation checklist scores than SIBO− equivalents. SIBO was significantly associated with worse symptoms of autism, which indicates that SIBO in children may significantly contribute to the symptoms of autism. Strategies to treat SIBO or to improve the gut microflora profile through dietary modulation may help to alleviate common gut disorders in children with autism [198].

**Group 9 summary:** Although represented only by ASD, this group occupies an important place in the review, as this disorder is becoming more and more common in the human population (prevalence of SIBO, 31%) [6]. Neurologists should pay attention to the fact that in SIBO+ patients, the symptoms of autism may become more severe, which may also be related to an inadequate diet.

### 5.10. Group 10: Mental Disorders

Attention has recently been paid to the correlation between depressed patients and intestinal microecology. Disturbances in the intestinal microecology lead to increased symptoms of anxiety and depression, which prevents proper functioning of the digestive tract, thus contributing to SIBO. Consequently, it is crucial to find a reliable and safe treatment for SIBO combined with depression [199].

SIBO syndrome has been associated with depression, stress, and anxiety [200,201,202]. Dysbiosis and inflammation of the gut have been linked to several mental health issues, including anxiety and depression, in children and adults. In particular, 50 people participated in the study, of which 26 were diagnosed SIBO+ and 24 SIBO− (non-SIBO control group). According to the conducted research, compared to the SIBO− subgroup, SIBO+ patients expressed specific patterns of personality traits, including higher neuroticism, lower extroversion, and a higher state of anxiety and stress [200]. Experimental studies have indicated that psychological stress can negatively affect the transit time of the small intestine, promote SIBO syndrome, and significantly disrupt the balance of the intestinal barrier [200]. Chronic activation of the hypothalamic–pituitary–adrenal axis may play a crucial role in developing SIBO, as the stress response is closely linked to the microecology of the gut [200]. Many researchers have focused on analyzing and assessing the mental state and gastrointestinal complaints of patients with SIBO with respect to tryptophan metabolism and rifaximin treatment [201,202]. SBO alters tryptophan metabolism, which can cause abdominal and mood disorders. Mild and moderate anxiety, as well as mild depression, were diagnosed in all SIBO patients. Changes in tryptophan metabolism were also observed in SIBO+ patients (with an increase in the activity of the serotonin pathway of tryptophan metabolism). Rifaximin treatment in SIBO+ patients ameliorated their mood disorders and gastrointestinal ailments, underlined by changes in tryptophan metabolism [202]. Other studies have shown that SIBO occurs in patients with depression and diabetes, with the incidence being twice that in the general population [199]. As suggested by the results, a capsule containing complex lactic acid bacteria may be beneficial for SIBO patients with depression and diabetes. It may alleviate the symptoms of depression, improve immune function, and reduce levels of inflammatory factors and fasting plasma glucose, with fewer side effects and potent effects. This may be because *Lactobacillus* regulates the expression of genes related to glucose and lipid metabolism [199].

**Group 10 summary:** Group 10 found that SIBO accompanies mental health problems such as depression, stress, and anxiety in patients. This is due to the colonization of bacteria in the small intestine adversely changing the metabolism of tryptophan—a precursor of serotonin, known as the happiness hormone—thus leading to mental health disorders.

### 5.11. Group 11: Genetic Diseases

#### 5.11.1. Cystic Fibrosis (CF)

The scientific literature has reported a high prevalence of SIBO in patients with cystic fibrosis (CF). Based on animal studies, it has been observed that the incidence of SIBO in CF ranges from 31% to 56%. The explanation seems to be a mutation in the *CFTR* gene, leading to increased secretion of abnormal condensed intestinal mucus, with damage to Paneth cells further reducing small intestinal motility [203,204]. According to other studies, the prevalence of SIBO in CF was 31.6%. SIBO was associated with pancreas insufficiency and lower body mass index [205]. Another study has reported the prevalence of SIBO in CF as 40%, and CF levels did not differ between SIBO+ and SIBO− patients. However, the potential link with intestinal inflammation has not yet been studied. Gastrointestinal inflammation is a frequent finding in cystic fibrosis patients. Despite this fact, SIBO does not seem to be the major (or, at least, not the only) determinant of intestinal inflammation [205].

#### 5.11.2. Familial Mediterranean Fever (FMF)

Familial Mediterranean Fever (FMF) is a genetic disease (due to a *MEFV* gene mutation) that causes recurrent fevers and painful inflammation of the abdomen, chest, and joints [206]. Colchicine is a fat-soluble alkaloid which binds to β-tubulin, inhibiting neutrophil chemotaxis and reducing the expression of adhesion molecules. It prevents febrile attacks and is used for controlling inflammation in the context of FMF; however, 5–10% of FMF patients are colchicine non-responsive [206]. This is probably related to vasculitis, inflammatory bowel disease, or occult infections, which are factors that reduce the effectiveness of drugs. It has been shown that SIBO affects the responsiveness to colchicine and the clinical severity of patients affected by FMF. It can be assumed that impaired intestinal bacterial products of intestinal microbiota may act in patients with innate immunity hypersensitivity such as FMF or Crohn’s disease, accentuating the clinical manifestations of autoinflammatory diseases. On the other hand, it cannot be excluded that SIBO may reduce the absorption of colchicine, thus reducing its effectiveness [206].

**Group 11 summary:** In summary, it should be stated that knowledge on the relationship between SIBO and group 11 diseases requires continued expansion. The first established facts, such as the relationship between SIBO and CF, make it possible to observe patients for SIBO and may help in solving clinical problems. For elucidation of the association between SIBO and FMF, further research in a representative cohort of patients is undoubtedly required.

### 5.12. Group 12: Gastrointestinal Cancer

To date, the associations between SIBO and pancreatic cancer, cholangiocarcinoma [207], gastric and colorectal cancer (CRC) [208,209], esophagogastric cancer (OGC) [210], and hepatocellular carcinoma (HCC) [211] have been considered in patients. Liang et al. [208] have shown that in Chinese patients, SIBO is associated with gastrointestinal cancer. The preliminary study concluded that probiotic intervention combats SIBO in patients with this cancer and alleviates its symptoms. Deng et al. [209] have investigated the prevalence of SIBO in patients with CRC after surgical treatment and observed whether gastrointestinal symptoms may improve with rifaximin. A total of 43 post-operative CRC patients and 30 healthy individuals were tested, and it was shown that post-operative CRC patients are more likely to develop SIBO compared with healthy individuals. SIBO may aggravate digestive symptoms. Rifaximin improved the overall gastrointestinal symptoms, particularly diarrhea, in SIBO+ patients. In patients after OGC resection [210], it has been shown that SIBO may not exhibit specific clinical symptoms, thus making its clinical diagnosis even more difficult and increasing the need to determine appropriate guidelines for its assessment and treatment. TLR4 protein expression in pancreatic carcinoma and cholangiocarcinoma patients was significantly higher in SIBO+ than SIBO− patients [207]. The relationship between the incidence of SIBO in patients with HCC has also been investigated [211]. Studies have shown that the expression levels of TLR2 and TLR4 and the incidence of SIBO in HCC patients were significantly higher than in cirrhosis and healthy control groups. Moreover, the high expression levels of TLR2 and TLR4 in SIBO+ HCC patients may promote the development of HCC.

**Group 12 summary:** Knowledge on the relationships between SIBO and diseases in group 12 indicates the great importance of this condition in cancer of the gastrointestinal tract. Oncologists should pay special attention to this problem in people after surgical treatment of CRC, where the risk of developing SIBO is high, thus intensifying digestive symptoms. Additionally, SIBO may promote the development of HCC. Undoubtedly, the association of SIBO with other cancers requires further research.

**Table 1 biomedicines-12-01030-t001:** Summary of findings under each group.

Groups	Diseases	The Impact of SIBO on the Disease
Group 1	Irritable bowel syndrome (IBS)	SIBO co-occurs in 51.7% of IBS patients Patients with IBS and SIBO have more severe gastrointestinal symptoms and intestinal communities dominated by the genus Prevotella SIBO induces immune system activity in IBS
	Inflammatory bowel disease (IBD) Crohn’s disease (CD) Ulcerative colitis (UC)	SIBO in patients with CD is related with more severe disease Eradication of SIBO using rifaximin reduced symptoms of IBD In IBD patients, SIBO increases blood endotoxin, TLR2, and TLR4 levels, mediating body inflammation Patients with UC easily experience SIBO
	Celiac disease (CeD)	SIBO can affect gastrointestinal motility in CeD Eradication of SIBO using rifaximin caused remission of symptoms in patients unresponsive to treatment with a gluten-free diet The association between SIBO and CeD requires further research, as CeD has a complex pathogenesis; hence, only specific variants may correlate with SIBO
	Non-alcoholic fatty liver disease (NAFLD)	Research showed that patients with NAFLD have an increased incidence of SIBO SIBO+ children were more likely to have NAFLD In patients with NAFLD, SIBO increases the risk of fatty liver disease SIBO in patients with NAFLD may be a factor contributing to increased transaminase activity, hepatic steatosis, and progression of liver fibrosis SIBO may contribute to the development of NAFLD through increasing intestinal permeability
	Liver cirrhosis	SIBO occurs in more than half of patients with cirrhosis SIBO in cirrhosis is associated with hepatic encephalopathy SIBO patients mainly have bacteria from the Blautia genus, which can convert primary bile acids into secondary bile acids
	Pancreatitis	SIBO can complicate chronic pancreatitis and interfere with management The presence of SIBO in acute pancreatitis correlates with the severity of the disease
Group 2	Systemic sclerosis (SSc)	SIBO occurs at a 10-fold higher incidence in SSc patients than in healthy people SIBO increases the risk of diarrhea in SSc patients In patients with SIBO and SSc, there is a significantly higher relative abundance of *Bacteroides* spp. and *Uncl. Rickenellaceae* spp. and significantly lower relative abundance of *Uncle. Erysipelotrichacaea* spp. Antibiotic treatment can eradicate SIBO and improve gastrointestinal symptoms in SSc patients; moreover, probiotics with *Saccharomyces boulardii* relieve the symptoms of SIBO FC may be a helpful test in identifying the group of SSc patients at high risk for SIBO FC levels may be helpful to assess SIBO eradication in SSc patients
Group 3	Heart failure (HF)	Patients with HF have a more dysfunctional intestinal flora than healthy people In patients with HF, the exhaled concentrations of hydrogen have been related to HF severity
	Deep vein thrombosis (DVT)	SIBO increased levels of inflammatory factors and expression of TLR4 and may be a risk factor for DVT
	Coronary artery disease (CAD)	Patients with SIBO may produce a larger quantity of endogenous alcohol, which may be related to CAD
	Subclinical atherosclerosis	SIBO is associated with impaired human vitamin K and increased risk for the development of atherosclerotic disease Subclinical atheromatous plaques are more common in SIBO+ patients
Group 4	Diabetes	SIBO is associated with type I and type II diabetes, as well as with gestational diabetes mellitus Hyperglycemia, delayed gastric emptying, impaired gut motility, and diabetic neuropathy may contribute to the development of SIBO during diabetes The blood glucose content of women with GDM and SIBO is higher than that in pregnant women with GDM without SIBO
	Hyperlipidemia	SIBO may cause hyperlipidemia via enterohepatic circulation disturbance SIBO is present in 78.9% of patients with hyperlipidemia and 40%
	Obesity	Obesity is associated with an increased risk of SIBO In obese patients, intestinal transit is slower There are more SCFAs in the large intestines of obese individuals
Group 5	Thyroid disease	More than 50% of patients with hypothyroidism suffer from SIBO Levothyroxine supplementation strongly affects the development of SIBO Other important factors for the development of SIBO in patients with hypothyroidism include impairment of intestinal clearance and immunosuppression Bacterial antigens can induce both Hashimoto’s disease and Graves-Basedow’s disease through modifying the immune response SIBO may affect gastrointestinal symptoms in patients with hypothyroidism
Group 6	Chronic kidney injury (CKI)	Renal failure predisposes individuals to changes in the gastrointestinal microbiome and development of SIBO An increase in *Enterobacterales* and *Enterococcus* genus bacteria with a concomitant decrease in *Lactobacillus*, *Prevotella*, and/or *Bifidobacterium* occurs Increased intestinal permeability in patients with CKI may increase the permeability of bacterial toxins that can damage the kidneys
	Acute kidney injury (AKI)	Ischemia–reperfusion injury of the kidney initiates the release of inflammatory mediators and the expression of adhesion molecules during AKI, which can induce intestinal dysbiosis Intestinal dysbiosis is also thought to be a factor in developing IgA nephropathy
Group 7	Acne rosacea	The incidence of SIBO in acne rosacea patients is very high, amounting to 46% SIBO increases intestinal permeability, resulting in the transfer of bacteria and pro-inflammatory cytokines into the systemic circulation, causing skin inflammation Eradication of SIBO with antibiotics may result in the regression of cutaneous lesions
	Psoriasis	SIBO may be related to the occurrence of psoriasis The use of rifaximin with a prebiotic may affect the eradication of SIBO, which may reduce the occurrence of psoriatic lesions
Group 8	Alzheimer’s disease (AD)	The incidence of SIBO in AD patients amounts to 48% The gut microbiome regulates the immune system and the central nervous system, affecting the adequate production of neuroactive molecules An abnormal structure of the intestinal microbiota results in the production of toxic metabolites that penetrate the bloodstream to the brain, causing inflammation in the nervous system The stools of AD patients present less diverse microbiome and calprotectin levels with respect to controls
	Parkinson’s disease (PD)	The incidence of SIBO in PD patients amounts to 46% SIBO is associated with worse gastrointestinal ailments and worse motor function in PD patients SIBO can trigger a local inflammatory response, disrupting the integrity of the intestinal barrier, which can affect local inflammation in the brain and neuronal susceptibility to neurodegeneration Intestinal dysfunction caused by SIBO results in impaired absorption of levodopa SIBO may also influence drug inactivation through increased production of reactive oxygen species SIBO in PD may result in decreased lipid absorption and weight loss
	Multiple sclerosis (MS)	SIBO is highly prevalent in patients with multiple sclerosis (MS) in certain geographical regions. This observation requires further research
Group 9	Autism spectrum disorders (ASDs)	In people with autism spectrum disorders, there is a significant modification of the composition of the intestinal microflora, including an increase in the amount of *Bacteroides* and *Clostridium perfringens* In autistic children, the incidence of SIBO is 31% SIBO is significantly associated with worse autism symptoms Strategies to treat SIBO may help alleviate common gut disorders in children with autism
Group 10	Mental disorders depression stress anxiety	Disturbances in the intestinal microecology lead to increased symptoms of anxiety and depression SIBO+ patients express specific patterns of personality traits: higher neuroticism, lower extroversion, and a higher state of anxiety and stress SIBO alters tryptophan metabolism, which can cause abdominal and mood disorders Rifaximin treatment of SIBO+ patients ameliorated their mood disorders and gastrointestinal ailments Administering a probiotic with lactic acid bacteria to patients with SIBO may alleviate symptoms of depression, improve the functioning of the immune system, and reduce the level of inflammatory factors
Group 11	Cystic fibrosis (CF)	The incidence of SIBO in CF patients may be as high as 50% Mutation in the CFTR gene and damage to Paneth cells limits the motility of the small intestine
	Familial Mediterranean Fever (FMF)	SIBO affects the responsiveness to colchicine and the clinical severity of patients affected by FMF. This observation requires further research
Group 12	Gastrointestinal cancer	SIBO is associated with gastrointestinal cancer Post-operative gastric and colorectal cancer patients are more likely to develop SIBO Probiotic intervention combats SIBO in patients with gastrointestinal cancer and alleviates its symptoms TLR4 protein expression in pancreatic carcinoma and cholangiocarcinoma patients was significantly higher in SIBO+ than SIBO− patients The expression levels of TLR2 and TLR4 and the incidence of SIBO in hepatocellular carcinoma (HCC) patients are significantly higher and may promote the development of HCC

Finally, the limitations in studying the relationships between SIBO and various diseases should be mentioned. The most important limitation is the methods used to test for SIBO, which need improvement and expansion. To overcome the shortcomings of current SIBO testing methods, new techniques are being developed based on molecular (e.g., the identification of different bacterial species based on 16S rRNA gene sequencing) and metagenomic approaches [5]. Another limitation is the difficulty in bringing together a representative cohort of patients from different geographical regions and different demographic structures, as well as representing different disease entities.

## 6. Conclusions

Awareness and knowledge about SIBO, as a relatively new disease, can change the lives of people affected by this problem for the better. So far, research on the relationships between SIBO and other diseases has focused mainly on diseases categorized into group 1 in this study, namely, gastrointestinal disorders, including irritable bowel syndrome, Crohn’s disease, ulcerative colitis, celiac disease, non-alcoholic fatty liver disease, liver cirrhosis, and pancreatitis. The associations of SIBO with diseases in this group have been well described, and SIBO may play an essential role in the pathogenesis of these diseases.

Through a comprehensive review, it was shown that an increasing body of evidence highlights the association of SIBO with other groups of diseases, including autoimmune, cardiovascular system, metabolic, endocrine, nephrological, dermatological, neurological, developmental, and mental disorders, as well as genetic diseases and gastrointestinal cancers. SIBO is a risk factor in many groups of diseases, complicating the course of these diseases and potentially playing a pathogenetic role in the development of their symptoms. In turn, metabolic diseases (e.g., diabetes) may be a predisposing factor for the development of SIBO. Knowledge regarding the associations between SIBO and various disease groups may help to provide better diagnoses and allow for the timely initiation of effective treatments or co-treatments. The current knowledge on SIBO certainly provides information that can be used to address various clinical difficulties throughout the course of SIBO-related diseases. On the other hand, it should be noted that the body of knowledge about SIBO certainly requires continued expansion.

## Figures and Tables

**Figure 1 biomedicines-12-01030-f001:**
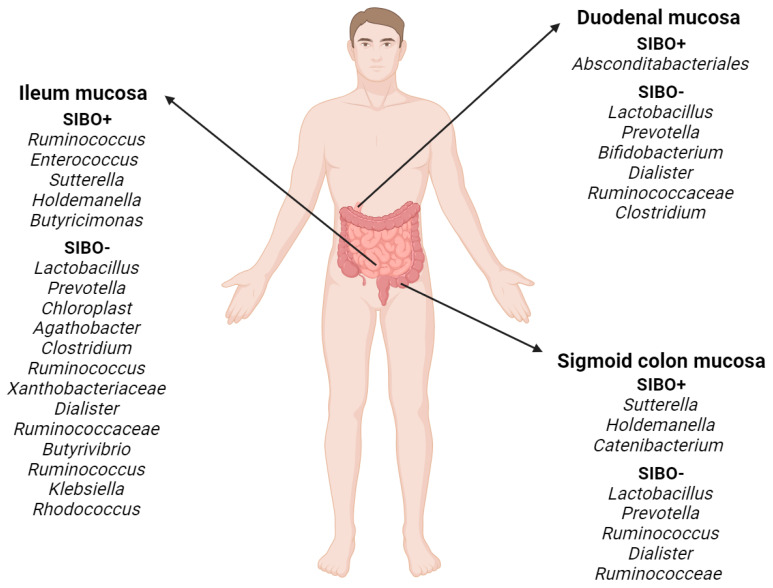
Mucosa-associated bacterial microbiota of multiple gut segments in people with SIBO (SIBO+) and healthy people (SIBO−). Developed based on [20].

**Figure 2 biomedicines-12-01030-f002:**
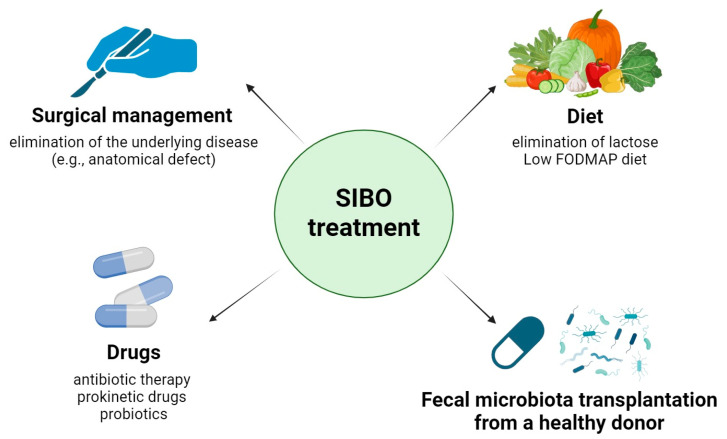
SIBO treatment methods.

**Figure 3 biomedicines-12-01030-f003:**
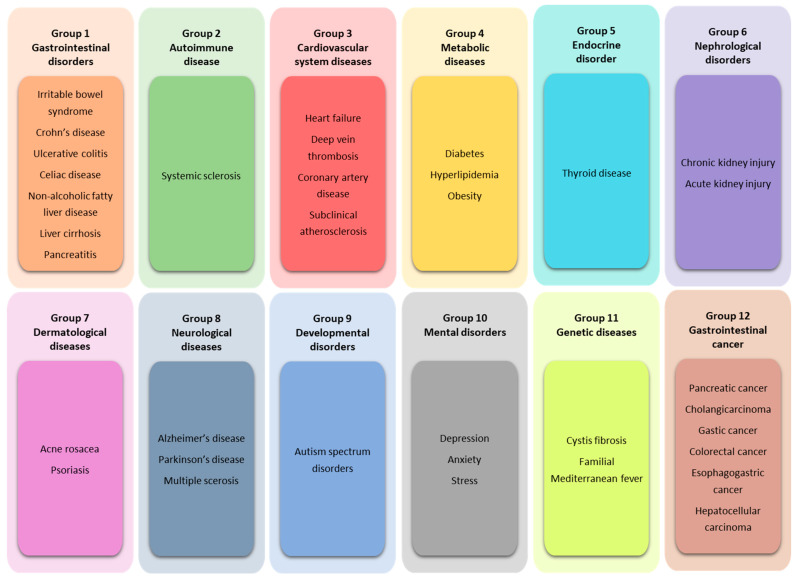
Twelve groups of diseases related to SIBO.

## Data Availability

Not applicable.

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
