# Peer review of "Small Intestinal Bacterial Overgrowth (SIBO) and Twelve Groups of Related Diseases—Current State of Knowledge"

_biomedicines, 2024, doi:10.3390/biomedicines12051030_

Round 1

Reviewer 1 Report

Comments and Suggestions for Authors

·         Abstract: (1) Most of these sentences should be better placed in introduction not in abstract (Line 23-31); (2) The process to literature searching is not clearly defined (e.g., PubMed, science direct, Scopus…etc.).

* English language should be improved. Please avoid using the word “we”, “our”, and “thanks to” (e.g., we divided, we can use, our comprehensive review).

·         Introduction: (1) The section is too short and not focused enough towards the research aim. Several references are missing where only 5 references were used; (2) clear description of the evidence gap that this review is filling is required. My main concern is that similar review papers have been performed. (Cureus. 2020 Jun; 12(6): e8860; World J Gastroenterol. 2023 Jun 14; 29(22): 3400–3421). No clear rationale on why this review is of importance. This should be clearly mentioned in the last paragraph.

·         Methods: It is advisable to include a methodology section, specifying the time frame used in the review, the database employed, the search terms, the study types (in vivo/vitro, RCT, quantitative, qualitative...etc.), and the inclusion/exclusion criteria for the reviewed articles. Incorporating these key aspects will enable a connection between the findings presented in this review and future reviews on this topic that are likely to be conducted.

·         Figure 1 is just copy and paste from other source. It is suggested to organize a new figure.

·         I would suggest to expand section 3 on SIBO prevalence, clinical features, current and developing diagnostic tests, and treatment.

·         Description of studies is vague (e.g., previous studies), suggest to better define the study type (quantitative, quantitative...etc.).

·         It is highly recommended to include tables or figures summarizing the findings under each group. Figure 3 has no meaning.

·         It is also advisable to specify the limitations of this review. This should be in a separate section.

Comments on the Quality of English Language

Moderate language editing is required.

Reviewer 2 Report

Comments and Suggestions for Authors

The aim of this review is to discuss the current evidence on the role of small intestinal bacterial overgrowth in the onset as well as exacerbation of clinical signs of selected diseases divided into six groups. Although the topic is interesting and overall the review is well organized, often some concepts/definitions are repeated over and over again (e.g., breath test) or, on the contrary, are scattered throughout the text. Singular sections should be more concise, especially 4.1.4. and 4.2.1., with punchier sentences and more accurate descriptions of the studies. The use of summary tables is recommended to make reading more understandable. Authors should also use acronyms carefully. Once the full name is provided on first use, only the abbreviation must be used thereafter. The English style should be reviewed for correction of typos and syntax errors.

Minor comments.

Figure 1. Please amend the typo in “Ileum mucosa”.

Figure 3. I suggest deleting the acronyms in brackets.

Comments on the Quality of English Language

Moderate revision required

Round 2

Reviewer 1 Report

Comments and Suggestions for Authors

Dear Authors,

Most of my comments are not sufficiently addressed.

1. The term "we" is still used in the manuscript (Lines 183-191, 230-234, 742). I would also suggest consulting with an English language speaker to edit the whole manuscript. Several sentences are difficult to understand.

2. I haven't seen the methodology section is added. Please refer to my comment (point 4).

3. It is highly recommended to include tables or figures summarizing the findings under each group. A descriptive summary is not enough.

Comments on the Quality of English Language

Extensive language editing is required.

Author Response

Response to Reviewer 1 Comments

We sincerely thank the Reviewer for the next one helpful comments.

We took into account all the Reviewer comments and explained a certain oversight the Reviewer accused us of regarding the lack of “Methodology” (details below).

Below, we respond step by step to the comments in the review. We've also included changes to the manuscript (this time marked in green).

Most of my comments are not sufficiently addressed.

Point 1: The term "we" is still used in the manuscript (Lines 183-191, 230-234, 742). I would also suggest consulting with an English language speaker to edit the whole manuscript. Several sentences are difficult to understand.

Response 1: In accordance with the Reviewer recommendation, the article was linguistically corrected by MDPI English Language Editing Services —the certificate is at the end of the document or at the Editor in cover letter (if the certificate cannot be sent by system).

 Point 2: 2. I haven't seen the methodology section is added. Please refer to my comment (point 4).

Response 2: Dear Reviewer, I am afraid there has been an oversight. We have included the „Methodology section” in the manuscript as recommended in the previously submitted version. After all, in point 1 of the comments addressed to us, the reviewer pointed out that we use the term "we" in lines 183-191, and this is the "Methodology" chapter. That's why I'm trying to understand where this situation comes from. We have fulfilled the request, as instructed by the Reviewer (line 188-197 in the current manuscript). I kindly ask you to accept the explanation.

Point 3: 3. It is highly recommended to include tables or figures summarizing the findings under each group. A descriptive summary is not enough.

Response 3: Following the Reviewer recommendation, we add Table 1 (location-Line: 1103-1107) presenting a summary of findings within each group. We also maintain descriptive summaries under each group.

The location of Table 1 is given in Section  5 (Line:346-352):

  1. SIBO-related diseases (Line:346-352)

The symptoms of SIBO may be limited to the gastrointestinal tract; however, a growing number of hypotheses have emphasized the association of SIBO with other diseases. The diseases retrieved during the literature review were divided into 12 groups (Figure 3). Diseases were described within the groups. A descriptive summary is provided under each group. In addition, Table 1 (at the end of Section 5 ) presents a summarizing the findings under each group.

Comments on the Quality of English Language: Extensive language editing is required.

Response : In accordance with the Reviewer recommendation, the article was linguistically corrected by MDPI English Language Editing Services. The certificate at the end of the document or at the Editor in cover Letter (if the certificate cannot be sent by system).

I would like to ask the Reviewer to accept our explanations and changes made to the manuscript and consent to publication in Biomedicines.

Best regards,

Beata Hukowska-Szematowicz

Reviewer 2 Report

Comments and Suggestions for Authors

I appreciate the efforts of the authors to address my previous comments and the paper is significantly improved. I only have some minor comments.

Line 182. Please amend the typo in Methodology.

Lines 233-234. The full names of GBT and LBT were previously reported on line 186.

Lines 261-262. This concept was already reported on lines 140-143.

Lines 462. NAFLD, not NAFL.

Line 543. Please amend the typo in NAFLD.

Comments on the Quality of English Language

Minor editing of English language required

Author Response

Response to Reviewer 2 Comments

We sincerely thank the Reviewer for the next one helpful comments.

We have introduced changes following the Reviewer suggestions.

We hope that with these changes, the manuscript will be acceptable for publication in Biomedicines.

Below, we provide a step-by-step response to the comments in the review. We also put the changes in the manuscript (marked in green).

I appreciate the efforts of the authors to address my previous comments and the paper is significantly improved. I only have some minor comments.

Point 1: Line 182. Please amend the typo in Methodology.

Response 1: I am very sorry for this oversight. Corrected.

Point 2: Lines 233-234. The full names of GBT and LBT were previously reported on line 186.

Response 2: Of course, corrected.

Point 3:
Lines 261-262. This concept was already reported on lines 140-143.

Response 3: In accordance with the Reviewer recommendation, sentence was corrected.

Point 4: Lines 462. NAFLD, not NAFL.

Response 4: Corrected.

Point 5: Line 543. Please amend the typo in NAFLD.

Response 5: Corrected.

Comments on the Quality of English Language: Minor editing of English language required

Response: In accordance with the Reviewer recommendation, the article was linguistically corrected by MDPI English Language Editing Services —the certificate is at the end of the document or at the Editor in cover letter (if the certificate cannot be sent by system).

I would like to ask the Reviewer to accept our explanations and changes made to the manuscript and

consent to publication in Biomedicines.

Best regards,

Beata Hukowska-Szematowicz

Round 3

Reviewer 1 Report

Comments and Suggestions for Authors

No further comments. The paper has significantly improved by these revisions.